# Orbital actinomadura madurae actinomycetoma: Case report and literature review

**Omayma Abdulazeem Abbass[1], Ali Awad Ali Hassan[2], Radwan Hassan Ali Hamed[3], Ali Awadallah Saeed[4,5]\*, Ahmed Hassan Fahal[5]**

**1** Faculty of Medicine, University of Bahri, Khartoum, Sudan, **2** Faculty of Medicine, Alribat University Hospital, Khartoum, Sudan, **3** Department of Medicine, Merowe Medical City, Merowe, Sudan, **4** Department of Pharmacology, Faculty of Pharmacy, The National University-Sudan, Khartoum, Sudan, **5** The Mycetoma Research Center, University of Khartoum, Khartoum, Sudan

\* alimhsd@gmail.com

## Abstract

The reported patient is a middle-aged farmer from West Sudan who presented with a painless mass in the right upper eyelid that progressively enlarged over time. His medical treatment journey was prolonged and difficult. The mass was initially mis-diagnosed as lacrimal gland pleomorphic adenoma. The patient was later referred to a specialised center, where wide local excision confirmed the diagnosis of *Actinomadura madurae* actinomycetoma. Although orbital mycetoma is uncommon, it is a serious condition that should be considered in the differential diagnosis in endemic areas.

## Author summary

We report a middle-aged farmer from West Sudan who had an orbital mycetoma. He had a mass in his right upper eyelid that was getting bigger but not hurting. The patient had a long and difficult medical journey because they were first given the wrong diagnosis of a lacrimal gland pleomorphic adenoma. It wasn't until a large local excision at a specialized center that the exact diagnosis of Actinomadura madurae actinomycetoma was made. Orbital mycetoma is a serious condition that should be thought about as a possible diagnosis, especially in places where it is common, to make sure that the right treatment is given at the right time.

## Introdusction

Mycetoma is recognised as one of the most neglected tropical diseases, which is a chronic, destructive, granulomatous inflammatory disorder caused by various micro-organisms, either fungal or bacterial, leading to eumycetoma or actinomycetoma,

provided the original author and source are credited.

**Data availability statement:** The authors declare that all data available in this manuscript. The submission contains all raw data required to replicate the results of your study.

**Funding:** The author(s) received no specific funding for this work.

**Competing interests:** The authors have declared that no competing interests exist.

respectively [1]. It spreads to the skin, deep tissues, and bones, causing tissue damage, deformities, disability, and potentially lethal consequences [2]. The orbital location is rare, and a serious condition that should be thought about as a possible diagnosis, especially in places where it is common, to make sure that the right treatment is given at the right time.

## Case presentation

The reported patient is a 38-year-old Sudanese farmer from El Obeid, Northern Kordofan State, in Western Sudan. His condition started in March 2023 when he noted a painless swelling in the right upper eyelid, mainly in the Superolateral portion of orbit. The swelling gradually increased in size and became fixed to the skin. Then, a skin sinus developed and started to drain a yellow discharge. He used different eye drops and antibiotics, but they did not improve his condition. In October 2023, he developed eye protrusion, drooping of the upper eyelid, eye redness and watering. Due to the ongoing conflict in the country, he moved to Northern Sudan, where he underwent several investigations that were consistent with the lacrimal gland pleomorphic adenoma. Surgical excision of the mass was attempted, but that was not successful due to a lack of proper ophthalmic surgical equipment, and only an incisional biopsy was taken. The histopathological diagnosis was lacrimal gland pleomorphic adenoma.

In October 2024, the patient was referred to the Ophthalmic Clinic at Merowe Medical City, Northern Sudan, for further management. At presentation, his medical history was unremarkable. He has neither a family mycetoma nor local trauma at the lesion. He is a smoker, and his past medical, social or geographical history was not contributory to his condition.

On examination, his general condition was good; he was oriented and not in pain. The systemic examinations were within normal. A local right eye examination revealed a palpable mass in the right upper eyelid, mainly in the superiolateral infrabrow area. The skin was dark in colour, with multiple closed small sinuses, with no discharge, and a horizontal scar. The mass was 9x4 mm in diameter, not tender, no local rise of temperature, with no thrill. It was attached to the skin and deep structures, firm in consistency, with mechanical ptosis. There was proptosis 24mm with inferonasal dystopia, injected conjunctiva, mild corneal abrasion, reactive pupil, normal ocular motility except limited right-up gaze movement and normal eye fundoscopy. The left eye examination was unremarkable. The visual acuity was normal in both eyes (Fig 1).

His complete blood count, hepatic and renal function tests, HIV screening, blood glucose level, urine general examination, and chest X-ray were unremarkable. He underwent a brain and orbit CT scan with contrast that showed a right superiolateral soft tissue lesion with enhancement not separable from the lacrimal gland, adhesion to the upper eyelid, and encasement of the superior-lateral eye globe (mass size = 2.5 × 2 cm). The orbital bony wall was intact with no bone molding, erosions, sclerosis, or destruction (Fig 2). The differential diagnosis included lacrimal gland pleomorphic adenoma and chronic inflammatory granuloma.

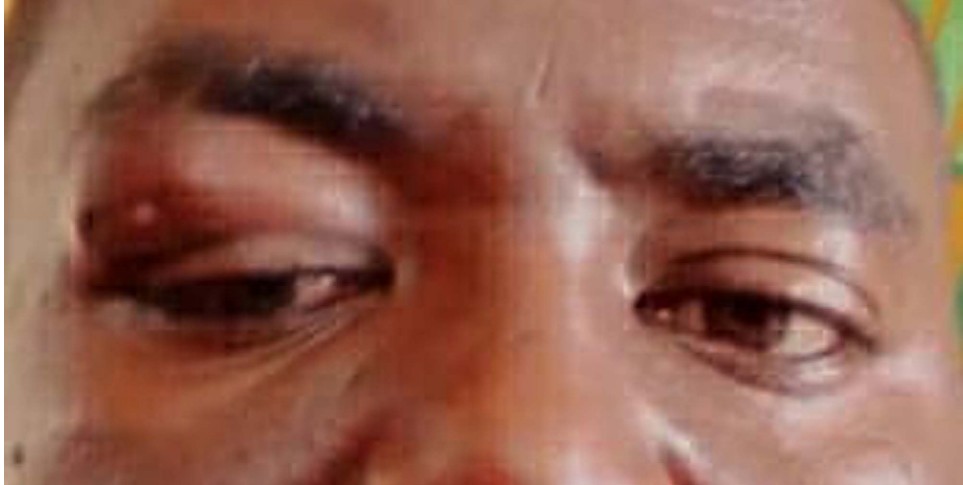

**Fig 1. Showing the pre-operative mass in the right upper eyelid, mainly in the superiolateral infrabrow area.**

He underwent wide local excision of the mass through lateral orbitotomy under general anaesthesia. Modified combined incisions were done, including a lazy-S incision in the upper eyelid crease, which allowed for complete excision of the mass with its muscular and skin attachments and maintained a reasonable skin for eyelid reconstruction (Fig 3). The tissues were dissected down to the periosteum. The periosteum was then incised over the orbital rim and extended two cm laterally along the Zygoma toward the Zygomatic Arch. Periosteal relaxing incisions were then made to allow posterior retraction of the temporal muscle and fascia. Then, the Temporalis Muscle and periosteum were elevated off the bone to expose the orbital rim and Zygoma externally. The periorbital tissue was elevated off the internal aspect of the lateral orbital wall, the lacrimal fossa, and the superolateral orbit as far back as the inferior orbital fissure. The intraorbital exploration and dissection were done, followed by excision to the lacrimal gland with the abnormal skin and orbicularis oculi muscle. The skin incisions were then extended nasally to maintain blepharoplasty, and reconstruction was done in layers. He had an uneventful postoperative recovery (Fig 3).

The surgical biopsy histopathological examination revealed two large grain fragments that have the characteristic multi-coloured pattern. The periphery of the fragments is dense, homogenous, and deep purple, while the centre is less dense, which is in line with *Actinomadura madurae.* The grain fragments are surrounded by a layer of polymorphonuclear leukocytes, with neutrophils closely attached to the surface of the grain and invading its substance. This is surrounded by a layer of cells consisting of plasma cells and macrophages (Fig 4).

He was started on a combination of amoxicillin and clavulanic acid in a dose of 1000mg/62.5mg twice daily and four doses of Co-Trimoxazole 80 mg/400 mg daily and to continue for six months to avoid recurrence. He underwent a postoperative brain CT scan, which was not remarkable, with no evidence of residual mass. The patient is on follow-up now with no drug adverse event.

## Discussion

The reported patient case underscores the substantial diagnostic challenges posed by mycetoma, especially in conflict-affected regions with limited healthcare resources. Due to instability in the area, the patient was compelled to seek treatment at various centers and hospitals, each with differing levels of expertise, diagnostic capabilities, and resources. These challenges are further compounded by the scarcity of specialised laboratories and trained personnel, which are essential for accurate identification and differentiation of mycetoma from other similar infections.

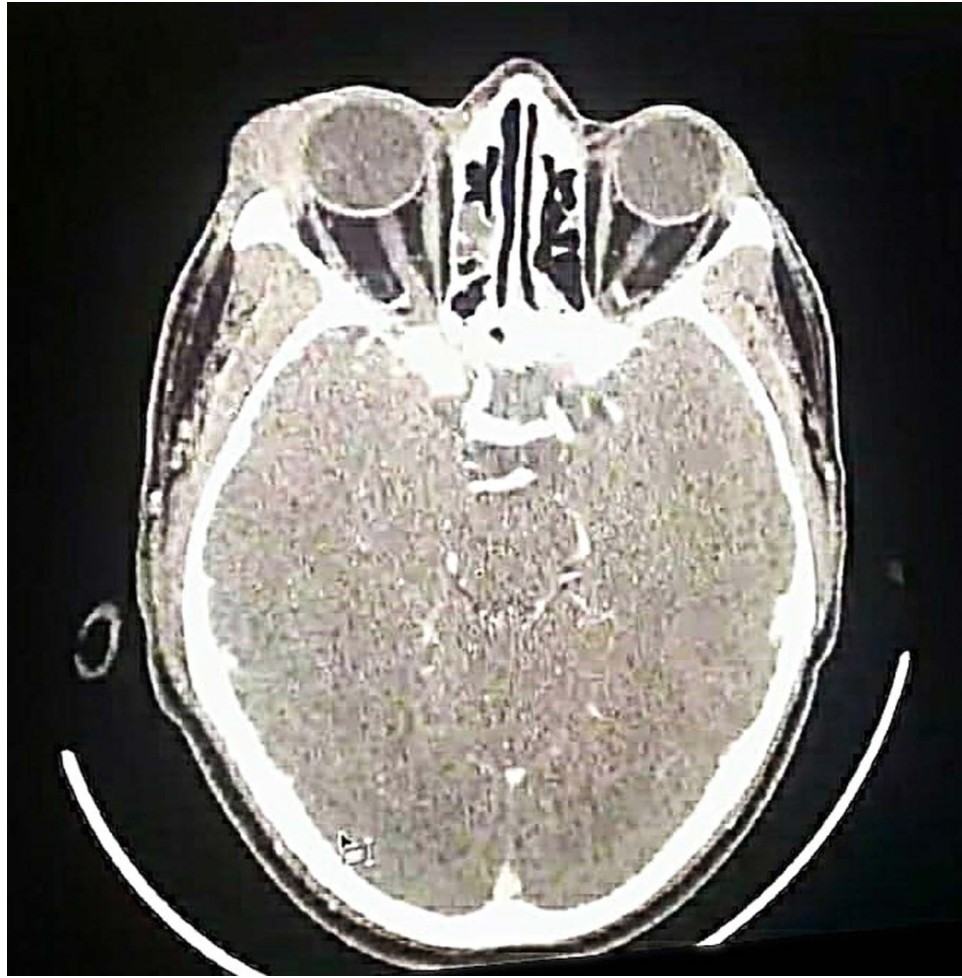

**Fig 2. Skull CT scan showing a right external canthus soft tissue lesion with enhancement not separable from the lacrimal gland, adhesion to the upper eyelid, and encasement of the superior-lateral eye globe (mass size = 2.5 cm).**

Mycetoma is a chronic, destructive, granulomatous inflammatory disease caused by numerous microorganisms, either fungal or bacterial, that lead to eumycetoma or actinomycetoma, respectively [1]. This deep subcutaneous mycosis spreads to the skin, deep tissues, and bones, leading to tissue damage, deformities, disability, and even death [2]. It has severe negative impacts on patients, communities, and healthcare systems in endemic regions [3]. In Sudan, eumycetoma is more prevalent, accounting for 70% of the reported cases, and *Madurella mycetomatis* is the predominant causative microorganism [4]. Actinomycetoma is less commonly encountered and frequently caused by *Streptomyces somaliensis* and less often caused by *Actinomadura madurae* [4].

Mycetoma often presents as a painless subcutaneous mass, accompanied by multiple sinus tracts with sero-purulent and purulent discharge containing grains [5]. However, clinical manifestations can be highly variable, making diagnosis challenging, especially in non-endemic areas. Delayed diagnosis can result in extensive tissue damage, disability and loss of function [6]. Disease progression and treatment outcomes vary depending on the causative organism, disease site and severity, and patient factors such as immune status and socio-economic conditions [7].

The extremities are affected most, accounting for 80% of the reported patients in Sudan [4]. Extrapaedal mycetoma is more frequently observed in the gluteal region, perineum, and back, and mycetoma affecting the head and neck is rare,

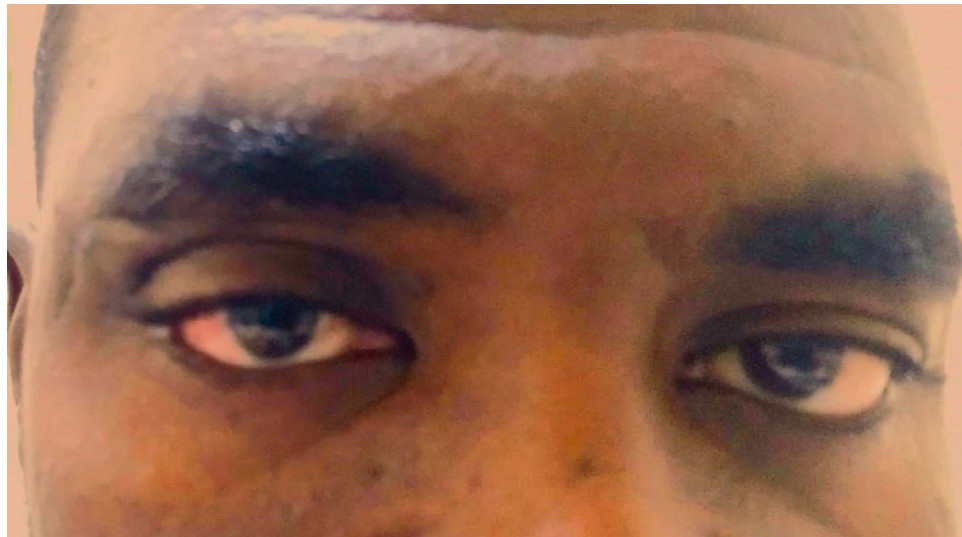

**Fig 3. Showing a healed post-operative incision in the right upper eyelid.**

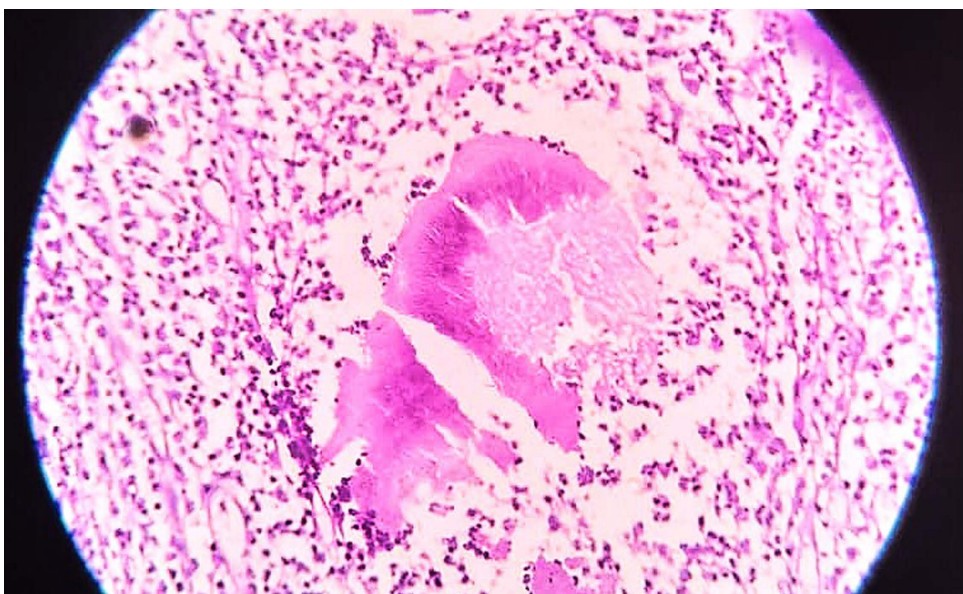

**Fig 4. Showing the surgical biopsy histopathological examination with evidence of *Actinomadura madurae* with Type I tissue reaction.** H&E X40.

with eye involvement being even more uncommon [8–10]. In the reported patient, the cause of the mycetoma is unclear, as there was no history of local trauma at the mycetoma lesion. The route of infection may be subcutaneous inoclusion through unnoticed minor skin trauma. Hence, the diagnosis was a histopathological surprise in the reported patient.

A literature review revealed only 16 documented patients with orbital mycetoma, and most of them were reported from Sudan, Table 1. The Mycetoma Research Center in 2014 reported on three patients with eumycetoma caused by *Madurella mycetomatis.* All patients had recurrent progressive left eye eumycetoma. The sites were the left upper eyelid,

**Table 1. The reported orbital mycetoma patients.**

| Date | Country | Patients No. | Type | Site | Treatment | Diagnostic method/ treatment outcome | Reference |
|------|---------|--------------|------|------|-----------|--------------------------------------|-----------|
| 2022 | Sudan | 1 | Actinomycetoma | • Orbit with cranial extension | Partial excision and medical treatment | – | [11] |
| 2018 | Espania | 1 | Eumycetoma | • Eyebrow | | – | [19] |
| 2017 | Canada | 1 | Scedosporium apiospermum eumycetoma | • Eyelid | voriconazole 200 mg po bid | Histopathology/complete response was observed after 18 weeks | [20] |
| 2014 | Sudan | 3 | *M. mycetomatis* eumycetoma | • left upper eyelid<br>• Left intraorbital<br>• Left temporal side of the caruncle. | Two had Surgical and all had Ketoconazole | Histopathology/The pain and discharge were reduced | [10] |
| 2012 | Senegal | 1 | *M. mycetomatis* Eumycetoma | • Eye | Surgical + medical (ketoconazole) | Histopathology/No improvement of vision | [14] |
| 2006 | South Korea | 5 | *Pseudallescheria boydii* eumycetoma | • Different parts of the eye were affected, and the conjunctiva was most frequently | Systemic and subconjunctival treatment with itraconazole | Microbiologic cultures/ Patient remainedsymptom free | [18] |
| 2010 | Hong Kong | 1 | *E dermatitidis* eumycetoma | • subconjunctival | Aggressive topical and systemic antifungal treatments following surgical intervention to surgical de´bridement alone. | 28S ribosomal RNA gene sequencing/ complete resolution of symptoms | [21] |
| 2010 | India | 1 | *Nocardia caused actinomycetoma* | • Scalp and left eyelid | Medical (streptomycin + rifampicin + cotrimoxazole) | Microbiology testing/ healing of sinuses and reduction in dystopia | [22] |
| 1964 | Sudan | 1 | *Streptomyces somaliensis* actinomycetoma | • Orbit | Several forms of antibiotic therapy | – | [12] |
| 1940 | Sudan | 1 | Mycetoma (unidentified causative organism) | • Eyelid | – | – | [13] |

intraorbital, and the temporal side of the caruncle. All patients presented late with a disease duration ranging between one to two years. They all received oral ketoconazole, and two had wide local surgical excision [10]. Recently, in 2022, a case of orbital actinomycetoma with cranial extension was reported from Sudan, and that patient had partial excision and medical treatment [11]. Lynch from Sudan in 1964 reported on 610 mycetoma patients, with only one patient with orbit actinomycetoma due to *Streptomyces somaliensis* [12]. In 1940, Aldridge and Kirk reported on a patient with eyelid mycetoma from Sudan; however, the causative organism was not identified [13].

From overseas, several cases of orbital mycetoma were reported. From Senegal, one patient with *M. myceto*matis eye eumycetoma was reported. He presented with progressive left proptosis of two years duration after local trauma. The clinical presentation was typical, and the diagnosis was confirmed by MRI and surgical biopsy histopathological examination [14].

Overseas five cases of ocular eumycetoma caused by *Pseudallescheria boydii* were reported. Different parts of the eye were affected, and the conjunctiva was most frequently involved, followed by the periocular region [15–18]. *Pseudallescheria boydii* infections have severe complications that can result in vision loss, making prompt and proactive management essential.

Furthermore, a case of a subcutaneous eumycetoma of the eyebrow in an immunocompromised patient was reported by Combalia and associates in 2018 [19]. A *Scedosporium apiospermum* eyelid mycetoma, masquerading as sebaceous carcinoma, was reported by Zoroquiain and colleagues in 2017 [20]. An immunocompetent woman presented with tears

containing black deposits that proved to be a presentation of subconjunctival eumycetoma was reported. The causative fungus was *E dermatitidis* [21]. A multiple actinomycetoma of the scalp and left eyelid was reported, and Nocardia was the causative microorganism [22].

Mycetoma in animals is an uncommon condition reported in a few cases. A painless subconjunctival mass involving the lateral canthus of the left eye in an American Saddlebred gelding was reported. The surgical biopsy, histopathological examination, and culture confirmed the diagnosis of *Scedosporium apiospermum* mycetoma [23].

Diagnosing mycetoma is a challenging and intricate process. Precise identification of the causative organism at the species level and assessment of the disease's extent across different tissue planes are essential for guiding appropriate treatment. This requires a combination of histopathological examination of surgical biopsies, grain culture, and molecular techniques to identify mycetoma-causing agents accurately [1,2]. Additionally, various imaging modalities are necessary to evaluate the spread of the disease along body planes, providing critical information for comprehensive disease management [1,3,5]. In low-resource mycetoma-endemic settings, these advanced diagnostic investigations are often unavailable, leaving many centers reliant on histopathological techniques to confirm the diagnosis.

A combination of medical treatment and surgical intervention remains a key approach in managing eumycetoma, especially when there is considerable mass effect or functional impairment. In this case, complete excision of the mass was achieved through lateral orbitotomy, a technique commonly used for orbital tumours. The surgery enabled the complete excision of the granuloma while preserving ocular function and maintaining aesthetic outcomes. In contrast, for actinomycetoma, medical treatment is preferred, as surgical intervention can cause more harm than benefit, as noted in a previous case report [11]. However, in the reported patient, the diagnosis of actinomycetoma was a surgical surprise.

Distinguishing between common eye fungal infections and eye eumycetoma is crucial, as their presentations and treatments differ significantly [24,25]. Eye fungal infections typically respond well to antimicrobial therapy, including antifungals and antibiotics for secondary bacterial infections. In contrast, treating eumycetoma often involves a combination of surgical excision and medical treatment.

## Conclusion

The reported patient case illustrates a complex diagnostic journey, marked by significant challenges that were further exacerbated by ongoing regional conflict. This instability forced the patient to seek treatment across multiple healthcare facilities, each with varying levels of resources and expertise. In low-resource settings, mycetoma can be particularly difficult to diagnose accurately due to limited access to specialised diagnostic tools, inconsistent availability of trained personnel, and often inadequate laboratory infrastructure. These constraints delay accurate diagnosis and, consequently, timely treatment, compounding the patient's suffering and the potential for disease progression. The patient experience underscores the critical need for improved diagnostic capabilities and continuity of care in conflict-affected and resource-limited areas, where patients with complex conditions like mycetoma are especially vulnerable to prolonged diagnostic delays and mismanagement.

## Ethical approval

Ethical approval was obtained from the National University, Ethical Committee, with reference number NU-REC/014-024/12. The patient gave written informed consent to report this study.

## Author contributions

**Conceptualization:** Omayma Abdulazeem Abbass, Ali Awad Ali Hassan, Radwan Hassan Ali Hamed.

**Data curation:** Ali Awadallah Saeed.

**Investigation:** Ali Awad Ali Hassan.

**Methodology:** Omayma Abdulazeem Abbass, Ali Awad Ali Hassan.

**Supervision:** Radwan Hassan Ali Hamed, Ahmed Hassan Fahal.

**Validation:** Ali Awadallah Saeed.

**Writing – original draft:** Radwan Hassan Ali Hamed, Ali Awadallah Saeed, Ahmed Hassan Fahal.

**Writing – review & editing:** Radwan Hassan Ali Hamed, Ali Awadallah Saeed, Ahmed Hassan Fahal.

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
