## [Decision Letter · Decision Letter 0]

3 Mar 2025

PNTD-D-24-01845Orbital Actinomadura madurae ActinomycetomaPLOS Neglected Tropical Diseases Dear Dr. Saeed, Thank you for submitting your manuscript to PLOS Neglected Tropical Diseases. After careful consideration, we feel that it has merit but does not fully meet PLOS Neglected Tropical Diseases's publication criteria as it currently stands. Therefore, we invite you to submit a revised version of the manuscript that addresses the points raised during the review process. Please submit your revised manuscript within 30 days May 01 2025 11:59PM. If you will need more time than this to complete your revisions, please reply to this message or contact the journal office at plosntds@plos.org. Please include the following items when submitting your revised manuscript: * A rebuttal letter that responds to each point raised by the editor and reviewer(s). You should upload this letter as a separate file labeled 'Response to Reviewers '. This file does not need to include responses to any formatting updates and technical items listed in the 'Journal Requirements' section below. * A marked-up copy of your manuscript that highlights changes made to the original version. You should upload this as a separate file labeled 'Revised Manuscript with Track Changes '. * An unmarked version of your revised paper without tracked changes. You should upload this as a separate file labeled 'Manuscript '. If you would like to make changes to your financial disclosure, competing interests statement, or data availability statement, please make these updates within the submission form at the time of resubmission. Guidelines for resubmitting your figure files are available below the reviewer comments at the end of this letter. We look forward to receiving your revised manuscript. Kind regards, Max Carlos Ramírez-Soto, BSc, MPH, PhD, FRSPHAcademic EditorPLOS Neglected Tropical Diseases Marcio RodriguesSection EditorPLOS Neglected Tropical Diseases

Shaden Kamhawi

co-Editor-in-Chief

Paul Brindley

co-Editor-in-Chief

**Additional Editor Comments :** Authors should revise the manuscript according to the reviewers' comments and the Editor's comments, and submit a revised version.

Comments:

1. Revise the title. For example. Orbital Actinomadura madurae Actinomycetoma: case report and literature review

2. Review the instructions to PLOSNTD authors, Research Articles should include an ‘Author summary’ section. Brief description of the case.

3. Include the CARE Checklist as supplementary material. In addition, you should ensure that you comply with each of its items.

4. Include an Introduction section with a brief paragraph from the literature review, including the objective of the case.

5. Include an Ethical Statement section. For example. This study was approved by the ethics committee of XXXXXXX and the study participant was informed about the study procedures and written informed consent was obtained.

6. Submit the original images in high quality to improve the quality of the report.

7. Case Report section. Describe briefly whether a differential diagnosis was considered in the case.

8. Table 1. Include the diagnostic method and treatment outcome. This will help a lot in practice.

**Journal Requirements:**

1) Please provide an Author Summary. This should appear in your manuscript between the Abstract (if applicable) and the Introduction, and should be 150-200 words long. The aim should be to make your findings accessible to a wide audience that includes both scientists and non-scientists. Sample summaries can be found on our website under Submission Guidelines:

2) Your manuscript is missing the following sections: Introduction, Results, and Methods.  Please ensure all required sections are present and in the correct order. Make sure section heading levels are clearly indicated in the manuscript text, and limit sub-sections to 3 heading levels. An outline of the required sections can be consulted in our submission guidelines here:

3) Thank you for including an Ethics Statement for your study. Please include:

i) The full name(s) of the Institutional Review Board(s) or Ethics Committee(s)

ii) The approval number(s), or a statement that approval was granted by the named board(s).

5) Please ensure that all Figure files have corresponding citations and legends within the manuscript. Currently, Figure 4 in your submission file inventory does not have an in-text citation. If the figure is no longer to be included as part of the submission, please remove it from the file inventory. Please note that Figure 4 which is included in the manuscript is different from the one uploaded in the online submission form.

Potential Copyright Issues:

i) Please confirm (a) that you are the photographer of 1, and 4, or (b) provide written permission from the photographer to publish the photo(s) under our CC BY 4.0 license.

ii) Figure 3. Please confirm whether you drew the images / clip-art within the figure panels by hand. If you did not draw the images, please provide (a) a link to the source of the images or icons and their license / terms of use; or (b) written permission from the copyright holder to publish the images or icons under our CC BY 4.0 license. Alternatively, you may replace the images with open source alternatives. See these open source resources you may use to replace images / clip-art:

7) We note that your Data Availability Statement is currently as follows: "The authors declare that all data available in this manuscript". Please confirm at this time whether or not your submission contains all raw data required to replicate the results of your study. Authors must share the “minimal data set” for their submission. PLOS defines the minimal data set to consist of the data required to replicate all study findings reported in the article, as well as related metadata and methods (https://journals.plos.org/plosone/s/data-availability#loc-minimal-data-set-definition).

8) Thank you for providing the written consent form. We notice that it is incomplete. Authors must obtain written consent from the patient or (if the patient is a child) the patient's parent/guardian and upload the form as an "Other" file with their submission. You can find the template consent form attached.

For submission guidelines on Case Studies, and a link to our template consent form, please visit: https://journals.plos.org/plosntds/s/other-article-types#loc-clinical-symposium.

**Reviewers' comments:**

Reviewer's Responses to Questions

**Key Review Criteria Required for Acceptance?**

**Methods**

-Are the objectives of the study clearly articulated with a clear testable hypothesis stated?

-Is the study design appropriate to address the stated objectives?

-Is the population clearly described and appropriate for the hypothesis being tested?

-Is the sample size sufficient to ensure adequate power to address the hypothesis being tested?

-Were correct statistical analysis used to support conclusions?

-Are there concerns about ethical or regulatory requirements being met?

Reviewer #1: It is a well-studied case with the correct methodology.

Reviewer #2: The objectives of the study is clearly articulated. No issues on the ethical or regulatory requirements.

**Results**

-Does the analysis presented match the analysis plan?

-Are the results clearly and completely presented?

-Are the figures (Tables, Images) of sufficient quality for clarity?

Reviewer #1: The identification result and response to treatment are adequate.

Reviewer #2: The data is well presentedbut figures quality is not clear in the PDF which I reviewed

**Conclusions**

-Are the conclusions supported by the data presented?

-Are the limitations of analysis clearly described?

-Do the authors discuss how these data can be helpful to advance our understanding of the topic under study?

-Is public health relevance addressed?

Reviewer #1: The conclusions to be reached are appropriate

Reviewer #2: Conclusions are supported by the data presented

**Editorial and Data Presentation Modifications?**

Reviewer #1: None

Reviewer #2: 1. In line number 24, affected region is mentioned as ‘superiolateral infra-brow’. ‘Superolateral portion of orbit’ may be used.

2. Yellow discharge in line no.26. To comment if grains/granules noted by patient.

3. In line number 37, what does family mycetoma refer to?

4. In line no.45, ‘at normal temperature’ can be changed to ‘no local rise of temperature’.

5. In line no. 52, ‘full blood count’ may be replaced by ‘complete blood counts.’

6. In line no. 57, what does ‘bone molding’ refer to? Sclerosis to be mentioned instead of sclerossi

7. The histopathological diagnosis does not mention size of grain fragment, or any further stains apart from H & E. Was any confirmatory test performed for Actinomadura madurae?

**Summary and General Comments**

Reviewer #1: This is an interesting case of periorbital A. madurae, it is well studied and above all well discussed.

I only suggest that you include something about the treatment of this etiologic agent with Ciprofloxacin and in particular with linezolid, which may be another good therapeutic option.

Bonifaz A, et al Update on actinomycetoma treatment: linezolid in the treatment of actinomycetomas due to Nocardia spp and Actinomadura madurae resistant to conventional treatments. Expert Rev Anti Infect Ther. 2025 Jan;23(1):79-89. doi: 10.1080/14787210.2024.2448723.

Reviewer #2: The manuscript is well written and worth publication in this journal

PLOS authors have the option to publish the peer review history of their article (what does this mean? ). If published, this will include your full peer review and any attached files.

**Do you want your identity to be public for this peer review?** For information about this choice, including consent withdrawal, please see our Privacy Policy .

Reviewer #1: No

Reviewer #2: **Yes: ** Shivaprakash M Rudramurthy

**Figure resubmission:** While revising your submission, please upload your figure files to the Preflight Analysis and Conversion Engine (PACE) digital diagnostic tool, https://pacev2.apexcovantage.com/. PACE helps ensure that figures meet PLOS requirements. To use PACE, you must first register as a user. Registration is free. Then, login and navigate to the UPLOAD tab, where you will find detailed instructions on how to use the tool. If you encounter any issues or have any questions when using PACE, please email PLOS at figures@plos.org. Please note that Supporting Information files do not need this step. If there are other versions of figure files still present in your submission file inventory at resubmission, please replace them with the PACE-processed versions.
---

## [Editor Report · Decision Letter 1]

26 Mar 2025

PNTD-D-24-01845R1Orbital Actinomadura madurae Actinomycetoma: case report and literature reviewPLOS Neglected Tropical DiseasesDear Dr. Saeed, Thank you for submitting your manuscript to PLOS Neglected Tropical Diseases. After careful consideration, we feel that it has merit but does not fully meet PLOS Neglected Tropical Diseases's publication criteria as it currently stands. Therefore, we invite you to submit a revised version of the manuscript that addresses the points raised during the review process. Please submit your revised manuscript within 30 days Apr 25 2025 11:59PM. If you will need more time than this to complete your revisions, please reply to this message or contact the journal office at plosntds@plos.org. Please include the following items when submitting your revised manuscript: * A rebuttal letter that responds to each point raised by the editor and reviewer(s). You should upload this letter as a separate file labeled 'Response to Reviewers '. This file does not need to include responses to any formatting updates and technical items listed in the 'Journal Requirements' section below. * A marked-up copy of your manuscript that highlights changes made to the original version. You should upload this as a separate file labeled 'Revised Manuscript with Track Changes '. * An unmarked version of your revised paper without tracked changes. You should upload this as a separate file labeled 'Manuscript '. If you would like to make changes to your financial disclosure, competing interests statement, or data availability statement, please make these updates within the submission form at the time of resubmission. Guidelines for resubmitting your figure files are available below the reviewer comments at the end of this letter. We look forward to receiving your revised manuscript. Kind regards, Max Carlos Ramírez-Soto, BSc, MPH, PhD, FRSPH, FECMMhttps://orcid.org/0000-0003-0471-6746Academic EditorPLOS Neglected Tropical Diseases Marcio RodriguesSection EditorPLOS Neglected Tropical Diseases

Shaden Kamhawi

co-Editor-in-Chief

Paul Brindley

co-Editor-in-Chief

**Additional Editor Comments:** The authors have corrected the comments. There are some minor comments from reviewers who haven't reviewed it.

Editor. Table 1. Include the diagnostic method and treatment outcome of case reports. This will help a lot in practice.

Reviewer #1. This is an interesting case of periorbital A. madurae, it is well studied and above all well discussed.

I only suggest that you include something about the treatment of this etiologic agent with Ciprofloxacin and in particular with linezolid, which may be another good therapeutic option.

Bonifaz A, et al Update on actinomycetoma treatment: linezolid in the treatment of actinomycetomas due to Nocardia spp and Actinomadura madurae resistant to conventional treatments. Expert Rev Anti Infect Ther. 2025 Jan;23(1):79-89. doi: 10.1080/14787210.2024.2448723.

Reviewer #2.

1. In line number 24, affected region is mentioned as ‘superiolateral infra-brow’. ‘Superolateral portion of orbit’ may be used.

2. Yellow discharge in line no.26. To comment if grains/granules noted by patient.

3. In line number 37, what does family mycetoma refer to?

4. In line no.45, ‘at normal temperature’ can be changed to ‘no local rise of temperature’.

5. In line no. 52, ‘full blood count’ may be replaced by ‘complete blood counts.’

6. In line no. 57, what does ‘bone molding’ refer to? Sclerosis to be mentioned instead of sclerossi

7. The histopathological diagnosis does not mention size of grain fragment, or any further stains apart from H & E. Was any confirmatory test performed for Actinomadura madurae?**Reviewers' comments:****Figure resubmission:** While revising your submission, please upload your figure files to the Preflight Analysis and Conversion Engine (PACE) digital diagnostic tool, https://pacev2.apexcovantage.com/. PACE helps ensure that figures meet PLOS requirements. To use PACE, you must first register as a user. Registration is free. Then, login and navigate to the UPLOAD tab, where you will find detailed instructions on how to use the tool. If you encounter any issues or have any questions when using PACE, please email PLOS at figures@plos.org. Please note that Supporting Information files do not need this step. If there are other versions of figure files still present in your submission file inventory at resubmission, please replace them with the PACE-processed versions. **Reproducibility:** To enhance the reproducibility of your results, we recommend that authors of applicable studies deposit laboratory protocols in protocols.io, where a protocol can be assigned its own identifier (DOI) such that it can be cited independently in the future. Additionally, PLOS ONE offers an option to publish peer-reviewed clinical study protocols. Read more information on sharing protocols at https://plos.org/protocols?utm_medium=editorial-email&utm_source=authorletters&utm_campaign=protocols

---

## [Editor Report · Decision Letter 2]

4 Apr 2025

Dear Dr Saeed,

We are pleased to inform you that your manuscript 'Orbital Actinomadura madurae Actinomycetoma: case report and literature review' has been provisionally accepted for publication in PLOS Neglected Tropical Diseases.

Best regards,

Max Carlos Ramírez-Soto, BSc, MPH, PhD, FRSPH, FECMM

Academic Editor

Marcio Rodrigues

Section Editor

Shaden Kamhawi

co-Editor-in-Chief

Paul Brindley

co-Editor-in-Chief

No comments

---

## [Editor Report · Acceptance letter]

Dear Dr Saeed,

We are delighted to inform you that your manuscript, "Orbital Actinomadura madurae Actinomycetoma: case report and literature review," has been formally accepted for publication in PLOS Neglected Tropical Diseases.

Best regards,

Shaden Kamhawi

co-Editor-in-Chief

Paul Brindley

co-Editor-in-Chief
